# Don’t Throw Away Your Notebook: Effects of Task Difficulty and Presentation Medium on Memory Performance

**DOI:** 10.3390/bs14100917

**Published:** 2024-10-09

**Authors:** Meng Han, Miaomiao Xie, Bingcan Li

**Affiliations:** 1School of Psychological and Cognitive Sciences, Peking University, Beijing 100871, China; 2College of Management and Economics, Shenzhen University, Shenzhen 518060, China

**Keywords:** memory, task difficulty, presentation medium, screen-inferiority effect, smartphone, electrical devices

## Abstract

People are increasingly reliant on various electrical devices for learning and memory, yet the implications and consequences of this dependence remain poorly understood. The present study aimed to investigate how learning through electrical media impacts recall under varying task difficulties. During this study, participants encoded information related to daily life situations (low difficulty), academic conceptual knowledge (middle difficulty), or associative word pairs (high difficulty), presented on smartphones, computers, or paper. At test, they recalled the omitted content based on the provided cue information. A significant screen-inferiority effect was observed for both computers and smartphones. However, the impairment related to computers disappeared in the retrieval of daily life situations and academic conceptual knowledge, whereas the impairment associated with smartphones was consistently present across all tasks. These results suggest that memory performance is modulated by the interaction between the presentation medium and the specific demands of the task, highlighting a more pronounced screen-inferiority memory effect when the media restrict the depth of processing or when the memory tasks pose greater external challenges. A deeper understanding of these factors can guide the optimization of electrical devices to enhance human memory abilities and functions.

## 1. Introduction

In this era of technological advancement, electronic devices such as smartphones and computers have become an indispensable part of our daily lives, a trend particularly evident in educational settings [1,2,3]. This trend has been further accelerated by the surge in online learning spurred by the COVID-19 pandemic (https://static.od-cdn.com/Special_Report.pdf, accessed on 22 May 2022). The growing reliance on electronic devices for learning prompts a hotly debated question: does the learning outcome, especially memory performance, remain consistent when the material is presented on digital screens as opposed to being printed on paper?

Recent meta-analyses indicate that readers frequently exhibit lower comprehension and memorization of text when it is presented on screens compared to when it is printed on paper [4,5,6,7]. This screen-inferiority effect has been extensively confirmed in memory retrieval tasks where participants read articles and then recall text details by free recall, cued recall, or objective recognition multiple-choice [8,9,10,11,12,13]. While this screen-inferiority effect may result from the screen characteristics, such as refresh rates, fluctuating luminance and contrast levels [14,15], and hence visual fatigue [10], it is primarily attributed to cognitive factors associated with a reduced depth of processing during learning [5,7]. Those factors include attenuated attention and working memory due to frequent engagement in media multitasking [16,17,18,19], superficial processing because of the habitual use of online processing strategies and browsing patterns [20,21,22], decreased availability of effective encoding cues—such as visual and tactile–kinesthetic cues [23,24]—and poorly calibrated metacognitive judgments [11,25,26].

Notably, the extent of the screen-inferiority effect may further differ drastically between specific electrical devices, such as smartphones versus computers [5], given their significant differences in characteristics which may affect the depth of processing. Certain factors specifically associated with smartphones may encourage memory and attention impairment [27,28]. For example, the small screens of smartphones can restrict text formatting, necessitate more scrolling, and hinder text navigation [5,29]. More importantly, smartphone multitasking has been suggested to cause significant distraction [16,30,31]. These distractions may arise even when there is merely a smartphone nearby [32], even if the smartphone has been turned off [33,34]. Finally, the belief that smartphones are primarily used for socializing and entertainment might lead to a reduction in self-regulation during learning [25]. However, it remains unclear how smartphones, as the presentation medium, influence memory, given the limited research directly using smartphones as the learning medium and the mixed results obtained. For example, Han et al. recently found that word recognition is better when learning in the smartphone situation compared to the newspaper situation, due to the popularization of mobile media [35]. In addition, there is little research directly comparing the memory effect on smartphones versus other devices, *although researchers have* begun to attempt to compare the information processing differences between these electrical devices [7]. Considering the potential detrimental factors of smartphones on cognitive processing, we hypothesize the following:

**Hypothesis** **1** **(H1).**
*Smartphones would exhibit a more pronounced screen-inferiority memory effect compared to computers.*


Nevertheless, screen interference in memory can disappear or even turn into mnemonic benefits when the cognitive demands of the memory tasks are relatively low [5]. In other words, the screen-inferiority effect is most often observed when the memory task requires deeper processing. For example, when learning expository texts, participants tend to remember more details from paper-based texts than those presented on screens [11,36,37], whereas the presentation format has minimal impact on memory and comprehension for narrative texts [38,39]. This disparity is attributed to the higher-level processing, complex academic vocabulary, and lessened connection to real-world situations that characterize expository texts, making them more challenging to recall and understand [40]. It is also possible that smartphones, sharing more characteristics that reduce the depth of processing, may be subject to less moderation by task demands. In other words, smartphones can induce a screen-inferiority effect on memory even when task demands are low, whereas these low task demands make the inferiority effect disappear for other media such as computers. Accordingly, we hypothesize the following:

**Hypothesis** **2** **(H2).***Task difficulty modulates the screen-inferiority effect on memory*. *Specifically, when task difficulty is high, the screen-inferiority effect exists for both smartphones and computers; however, when task difficulty is low, the screen-inferiority effect only exists for smartphones*.

*In the current study, we tested these two hypotheses by conducting* three memory tasks, where the task demands were manipulated to be low (task one, involving daily life situations, corresponding to narratives), middle (task two, involving academic conceptual knowledge, corresponding to expository), or high (task three, eliminating genre-related information, but involving word pairs and refined spelling processing). From a practical perspective, these three tasks also align with three common scenarios in which electronic devices are frequently used: memorizing vocabularies, taking course notes, and keeping daily memos. Specifically, our study utilized the classical cued recall paradigm [41,42,43] where participants were asked to study individual pieces of information (instead of *entire passages*) and subsequently recall related information based on retrieval cues. This approach was chosen to ensure that we evaluated pure memory without the confounding factors from text reading and comprehension [44]. Finally, we compared two prevalent devices, computers and smartphones, in terms of their ubiquitous use in daily life but significant differences in physical characteristics, usage goals, and situations. By presenting memory materials across these two devices, compared to traditional paper, and manipulating the difficulty of memory tasks, we aim to shed light on how the presentation medium influences memory performance and the modulating effect of task demands.

## 2. Method

### 2.1. Participants

The sample size was determined through an a priori power analysis (MorePower 6.0) [45]. To obtain adequate power (α = 0.01, 1 − β > 0.80) on the basis of a moderate effect size (f = 0.25) for the interaction between task demands (with 3 levels) and presentation medium (with 3 levels), a minimum total of 46 participants per group were required for an ANOVA-based analysis strategy. However, considering the larger participant requirement of linear mixed modeling and the preference for a sample size larger than necessary to avoid insufficient power, we decided to increase the sample size by 20 participants to reach at least 66 participants per group.

A total of 215 students from Peking University participated in the experiment and received USD 3 compensation. They were healthy, right-handed native Chinese speakers, with normal or corrected-to-normal vision. None of them reported a history of neurological or psychiatric disorders, learning disorders, head injury, or psychotropic drug use. Informed consent was obtained from each participant. Data from five participants were discarded: two had prior experience with Swahili and three dropped out midway. Our final sample, therefore, included 210 participants (134 females; mean age = 20.20 years, *SD* = 2.34 years). They were randomly assigned to the smartphone group (*n* = 70; 40 females; mean age = 20.36 years, *SD* = 2.58 years), the computer group (*n* = 69; 43 females; mean age = 20.30 years, *SD* = 2.25 years), and the paper group (*n* = 71; 51 females; mean age = 19.93 years, *SD* = 2.13 years). This research was conducted in accordance with the ethical standards set forth in Declaration of Helsinki.

### 2.2. Stimuli

In the study, three types of material were used: sentences depicting daily life situations, sentences explaining academic conceptual knowledge, and Swahili–Chinese word pairs.

Nine sentences were created to portray a range of daily life scenarios, such as examinations, exercise, and reading. Each scenario consisted of four essential elements: time, location, the main character, and the specific event/action [46]. Furthermore, the scenarios were set in the past (3), present (3), or future (3), and involved characters who were either the individual themselves (3), an unrelated male (3), or an unrelated female (3). The emotional valence of each event was also varied, being negative (3), positive (3), or neutral (3). For example, “李明上周在咖啡馆读完了一本书”, meaning “Last week, Ming Li finished reading a book in the coffee shop”, depicted a neutral event that happened to Ming Li in the past. The total number of characters across all sentences was 148, and the average sentence length was 16.44 ± 3.21 characters.

In addition, nine sentences were selected from the encyclopedia to depict academic conceptual knowledge in the natural sciences, including topics such as geography, biology, and physics. All the knowledge points were unfamiliar to university students. An example was “前斜角肌的终点为第一肋骨上面斜角肌结节”, meaning “The end of the scalenus anterior is the scalene tubercle above the first rib”. The total number of characters across all sentences was 148, and the mean sentence length was 15.44 ± 4.48 characters.

Finally, 30 Swahili–Chinese word pairs which are frequently used in everyday life were included, covering topics like dates, designations, animals, and objects, etc., such as “upawa-勺子” (spoon). Some of the word pairs were selected and translated from Nelson and Dunlosky [47], and the remainder were translated by the authors of this report [48,49,50]. The mean word frequency for the Chinese words was 0.06 ± 0.11. The mean word length was 5.87 ± 1.96 characters for Swahili words, and 2.03 ± 0.72 characters for Chinese words [51].

### 2.3. Procedure

Participants entered a quiet experimental room, with the computer group and the smartphone group each carrying their private laptops and smartphones, separately. The experimental stimuli were presented on laptops (for the computer group), smartphones (for the smartphone group), or on A4 paper (for the paper group). Text was presented in size 4 Song font (in Times New Roman for Swahili words) with single-line spacing. The absolute text size and spacing would adjust proportionally and automatically according to the screen size of the electrical devices. Providing personal devices to participants was intended to mirror real-world scenarios of learning and memorization.

Participants were instructed to complete three memory tasks: learning and recalling sentences about daily life situations (task one), academic conceptual knowledge (task two), and Swahili–Chinese word pairs (task three). The task order was counterbalanced among participants. A standard study–test paradigm was employed for these tasks, with the study phase immediately followed by the test phase.

In task one and two, all sentences were presented on a single page or screen. During the study phase, participants were asked to memorize these sentences for a duration of two minutes. During the test phase, sentences were presented again with a key piece of information omitted and replaced by an empty space. For example, participants might be given the statement: “*The end of the scalenus anterior is ( )*”. They were then prompted to recall the missing details and complete the blanks based on the remaining information (given cues). Participants were allowed a maximum of three minutes to recall.

In task three, 30 Swahili–Chinese word pairs were divided into three sets, each containing 10 word pairs. Each set was presented on a separate page or screen. During the study phase, participants studied these pairs page by page (or screen by screen), with each page/screen displayed for three minutes. During the test phase, participants were required to recall the associated word and fill in the blanks based on the cued words. Half of the word pairs were cued with Chinese words, and the other half were cued with Swahili words. Participants had a maximum of five minutes for recall.

### 2.4. Scoring

For tasks one and two, a researcher who was not informed of the experimental purpose scored participants’ responses and a researcher from this report double-checked the scoring. Each task had a score of 90 points, with each blank worth 10 points. A rubric that propositionalized each blank into 1 to 4 idea units was used to code the recall data and the score for each idea unit was evenly distributed among the blanks it belonged to. For example, for the conceptual knowledge “The end of the scalenus anterior is ( )”, the required content consisted of four idea units, that is, “scalenus”, “ tubercle”, “above”, and “first rib”. Each idea unit was worth 2.5 points. Scoring was used to precisely assess how much detail each participant could recall rather than simple binary judgments of what was remembered or forgotten. For task three, responses were considered correct only when participants completely recalled and spelled the target word.

### 2.5. Statistical Analysis

All statistical analyses were conducted using R (v.4.0.2; R Core Team, 2020). Analyses for accuracy rates were conducted using linear mixed-effects models (LMM; function lmer, package lme4, v.1.1-27) [52], which can examine the effect of the presentation medium and its interaction with task demands by simultaneously modeling other potentially confounding factors and the variance associated with each subject. Categorical variables including presentation medium, task demands, and gender were sum-coded, and continuous variables including recall rates and age were z-transformed to help with model estimation and centered to make interactions interpretable and avoid nonessential multicollinearity [53,54].

We fitted the model that included the fixed effects for presentation medium, task demands, and their interaction, as well as age and gender as covariates. The random effect structures included by-subject intercepts for task demands. For statistical inferences about predictors, a fixed effect was considered significant if the p-values of the beta coefficients (obtained using type II Wald chi-square tests with the function ANOVA; car package for LMM: v.3.0-10) [55] were at *p* < 0.05. Post hoc tests for main effects and follow-up analyses for significant interactions were conducted using the *emmeans* package (v.1.6.1) [56], with Bonferroni corrections applied for multiple comparisons.

## 3. Results

The results of accuracy rates are visualized in Figure 1. Statistical analysis of the accuracy rates revealed a significant main effect of the presentation medium, χ^2^(2) = 31.52, *p* < 0.001, with lower accuracy rates for materials presented on smartphones compared to computers (β_diff_ = 0.34, *SE* = 0.10, 95% CI [0.10, 0.57], *t* (198) = 3.40, *p* = 0.002) and papers (β_diff_ = 0.52, *SE* = 0.10, 95% CI [0.28, 0.77], *t* (198) = 5.05, *p* < 0.001), but no significant difference between computers and papers, *t* (198) = 1.77, p = 0.18. A main effect was also revealed for task demands, χ^2^(2) = 739.22, *p* < 0.001, with lower accuracy rates for the retrieval of word pairs compared to life situations (β_diff_ = 1.23, *SE* = 0.06, 95% CI [1.09, 1.36], *t* (396) = 21.70, *p* < 0.001) and conceptual knowledge (β_diff_ = 1.28, *SE* = 0.06, 95% CI [1.15, 1.42], *t* (396) = 22.66, *p* < 0.001), but no significant difference between life situations and conceptual knowledge, *t* (396) = 0.96, *p* = 0.60. Additionally, there was a significant interaction between the presentation medium and task demands, χ^2^(4) = 10.43, *p* = 0.034.

Further follow-up pairwise comparisons revealed lower accuracy rates for smartphones compared to computers and paper when retrieving daily life situations (smartphone vs. computer, β_diff_ = 0.37, *SE* = 0.13, 95% CI [0.07, 0.66], *t* (432) = 2.90, *p* = 0.011; smartphone vs. paper, β_diff_ = 0.32, *SE* = 0.13, 95% CI [0.01, 0.63], *t* (432) = 2.46, *p* = 0.038; computer vs. paper, *t* (432) = 0.32, *p* = 0.95) and academic conceptual knowledge (smartphone vs. computer, β_diff_ = 0.39, *SE* = 0.13, 95% CI [0.10, 0.69], *t* (432) = 3.12, *p* = 0.006; smartphone vs. paper, β_diff_ = 0.59, *SE* = 0.13, 95% CI [0.28, 0.90], *t* (432) = 4.51, *p* < 0.001; computer vs. paper, *t* (432) = 1.50, *p* = 0.29). However, both smartphones and computers showed lower accuracy rates compared to paper when retrieving word pairs (smartphone vs. paper, β_diff_ = 0.65, *SE* = 0.13, 95% CI [0.34, 0.96], *t* (432) = 4.97, *p* < 0.001; computer vs. paper, β_diff_ = 0.40, *SE* = 0.13, 95% CI [0.09, 0.71], *t* (432) = 2.99, *p* = 0.008; smartphone vs. computer, *t* (432) = 2.02, *p* = 0.11).

Therefore, recall performance was impaired by smartphones and computers, while impairment with computers disappeared when participants retrieved daily life situations and academic conceptual knowledge. Moreover, computers demonstrated a memory advantage over smartphones in these two tasks.

## 4. Discussion

The present study employed a cued recall paradigm across three memory tasks to explore the impact of the presentation medium (computer vs. smartphone vs. paper) on the memorization of daily life situations, academic conceptual knowledge, and associative word pairs, with low, middle, and high difficulty levels, respectively. An inferiority effect was observed on computers when the retrieval content consisted of word pairs, with decreased memory performance for materials presented on computers compared to paper. However, this inferiority effect disappeared when recalling life situations and academic knowledge. In contrast, when materials were presented on smartphones, the inferiority effect was consistently present across all three tasks. Furthermore, memory performance was superior for computers compared to smartphones when retrieving life situations and academic knowledge, but there was no significant difference between those two devices in the retrieval of word pairs.

In keeping with previous studies showing that electrical devices as reading media can impair memory for text details—a phenomenon referred to as the screen-inferiority effect [9,10,11,12,13,57]—the present study also found lower accuracy rates for retrieving word pairs presented on computers compared to those presented on paper in a learning context. This memory disadvantage observed with computers might be caused by factors directly related to digital technology, such as screen-induced eye strain, visual fatigue and attentional interference [10,20,58], and/or cognitive factors associated with a reduced depth of processing, such as reduced sustained attention and working memory [17,18,19,59].

However, this computer-induced inferiority effect observed in retrieving word pairs diminished when retrieving life situations and academic knowledge, highlighting the modulating influence of task difficulty. In our study, retrieving word pairs involved not just individual items but also their associative relations [60,61,62] and precise spelling [48]. Those requirements increased the retrieval difficulty and magnified the screen-inferiority effect. Conversely, retrieving life situations and academic knowledge presented less challenges, thereby eliminating the inferiority effect. Accuracy rates further illustrate the greater difficulty in the retrieval of word pairs compared to life situations and conceptual knowledge. Note that although life situations and academic knowledge were manipulated to represent different levels of task difficulty, there was no significant difference in their memory performance in fact. This is because the present study used separate and unrelated pieces of information instead of long texts, which decreased the overall difficulty and diminished their differences. On the other hand, reduced task difficulty also leads to a ceiling effect (see Figure 1) for the retrieval of life situations and academic knowledge, which can present obstacles to the observation of differences between the paper and the computer, especially for the academic knowledge condition, which was anticipated to have a middle level of difficulty. In summary, our findings align with various perspectives suggesting that the impact of screen reading becomes more pronounced when tasks are inherently more challenging, such as reading under time pressure (vs. without time pressure) [63,64], reading longer texts (vs. shorter ones) [10,65], or recalling textual details (vs. understanding the text’s gist) [11,37].

Interestingly, when materials were presented on smartphones, the modulating effect of task difficulty, which was typically observed on computers, unexpectedly disappeared. Instead, a robust and stable screen-inferiority effect emerged across all three tasks, spotlighting the unique detrimental impact of smartphones on memory performance. Some factors particularly associated with smartphones have been suggested to impair memory performance [16,27,28,30,31]. Consistent with this research, the present study further suggests that the usage of smartphones may reduce the depth of processing and then impair memory performance, even for tasks that are less challenging. However, caution is warranted when drawing this conclusion, as task difficulty is a continuous variable. For example, recall of academic knowledge and life situations involves deeper processing compared to tasks that merely require the recognition of correct responses rather than their construction [66]. We suspect that the recognition task employed by Han et al. reduced task difficulty and consequently obscured the potential screen-inferiority effect [35]. Specifically, participants in their study were asked to judge whether the words in the test phase had previously appeared in the article that was presented on the smartphone or newspaper. Differently from tasks requiring participants to retrieve article details, the retrieval of single words does not necessitate any construction of meaning, making deep processing less necessary. Therefore, even smartphones with considerable constraints on cognitive resources can readily perform this task. In addition, the recognition of isolated words may even benefit from the online processing strategies and browsing patterns which are usually adopted when using smartphones. This could account for the observed LPC old/new effect (an ERP effect representing the recollection process and better memory) in the smartphone situation. In summary, these findings suggest that computers are more likely to reach the same level as paper, whereas smartphones can achieve similar levels only when task difficulty is reduced to a great extent. Future research should explore the turning point of task difficulty at which smartphones demonstrate enhanced performance.

The challenges associated with smartphones pop up again in the direct comparison of memory performance between smartphones and computers, with smartphones exhibiting lower accuracy rates compared to computers (directly supporting H1). Notably, this disparity was observed only in tasks involving life situations and academic knowledge, but not in the task involving word pairs (directly proving H2). This implies that the medium effect on memory performance also varies with task difficulty. Specifically, tasks that were overly challenging led to a shared constraint for both devices. However, in simpler tasks, distinctions became apparent, with computers showing a ceiling effect, whereas smartphones presented more obstacles that could potentially impair memory performance. Given that previous research on medium effect yielded mixed results [38,67,68], we suspect that it may be due to the insensitivity of the tasks employed to detect such differences. Our findings indicate a complex situation, underscoring that research on the medium effect needs to be informed by gradient data on task difficulty.

One major limitation of our study is the absence of individual difference measures, such as problematic smartphone use and social anxiety, which could have been used as covariates in our statistical analyses. Furthermore, this study included only three levels of difficulty, limiting our ability to capture the detailed trend that depicted the memory performance across task difficulty when using electronic devices. Finally, our focus was exclusively on accuracy rates, ignoring other critical indices that can reflect memory performance such as reaction times (RTs) and confidence levels in making responses. To address these issues, further research should extend the current work by incorporating the moderating impacts of various individual differences, by constructing task difficulty gradients, and by utilizing more measurement methodologies.

## 5. Conclusions

In summary, the present study revealed that the screen-inferiority effect on memory was modulated by the interaction between task demands and the presentation medium. Hence, we propose that in order to improve memory outcomes, the selection of the appropriate material presentation medium should be driven by the specific demands of the task. For instance, for challenging tasks such as vocabulary learning, traditional paper remains the best option. Moreover, computers can be used for simpler tasks such as taking notes and keeping reminders, whereas smartphones are better suited for capturing fragmented information, not for learning scenarios that require careful processing. On the other hand, to boost memory performance on electrical devices, especially smartphones, targeted optimizations are recommended, such as improving the screen, and, more importantly, constructing a focused usage environment. It is crucial to recognize that digital technologies are ultimately mere tools. Future research should explore how these tools can be designed and utilized to effectively enhance, rather than constrain, the abilities and functions of human memory.

## Figures and Tables

**Figure 1 behavsci-14-00917-f001:**
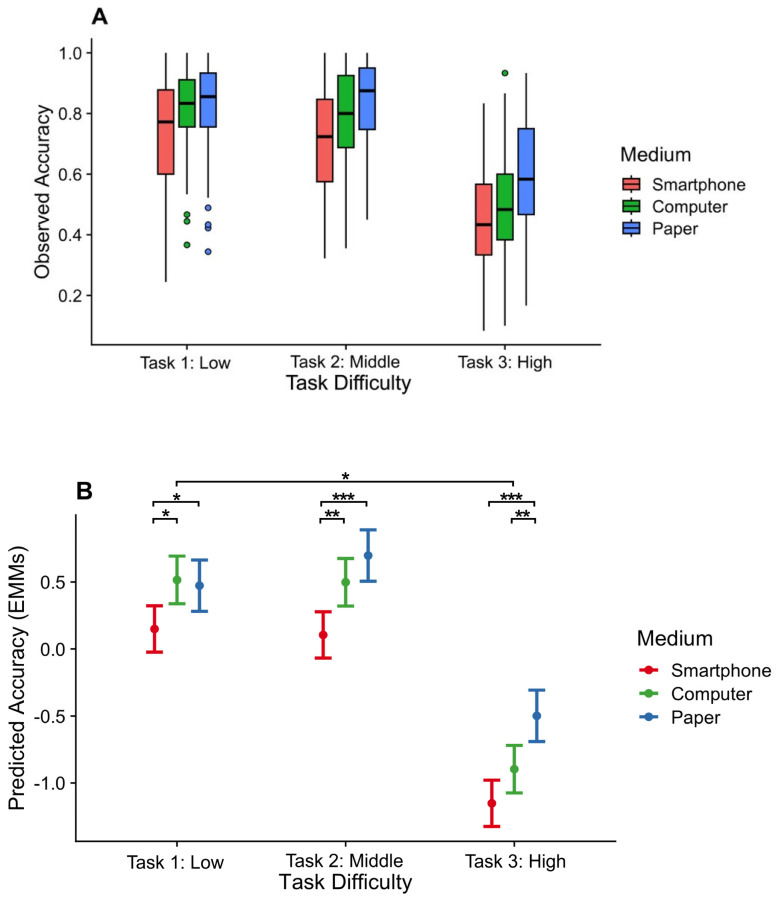
Observed accuracy rates (**A**) and predicted accuracy rates (**B**). *Note*. Accuracy rates were reduced by smartphones across all tasks, while they were only reduced by computers in Task 3, which had higher processing demands. Additionally, the accuracy rates for computers were higher than those for smartphones in Tasks 1 and 2, which had lower processing demands. The plots in (**A**) show boxplots based on raw data that were aggregated by participant. Upper whisker = largest observation less than or equal to upper hinge + 1.5 × IQR (IQR is the inter-quartile range); lower whisker = smallest observation greater than or equal to lower hinge − 1.5 × IQR. The points represent the outliers. The plots in (**B**) show the estimated marginal means for the fitted accuracy (z-transformed). Error bars indicate 95% CIs. *, *p* < 0.05; **, *p* < 0.01; ***, *p* < 0.001.

## Data Availability

The data are available from the corresponding author upon reasonable request.

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
