# Peer review of "Don’t Throw Away Your Notebook: Effects of Task Difficulty and Presentation Medium on Memory Performance"

_behavsci, 2024, doi:10.3390/bs14100917_

Round 1
Reviewer 1 Report
Comments and Suggestions for Authors
Overall, the paper is concise and well written. Yet, there are a few points I would like to see improved:
1. In reading section “2.5 Statistical analysis”, it was not clear to me, on first glance, whether task difficulty was treated as categorical variable and therefore sum coded, or whether the ordinal character of the variable was exploited in the analysis. I hope that the variable was treated as categorical, because obviously, the manipulation of the task difficulty did not work as it was supposed to. I would prefer if it was explicitly stated in this section which variables were treated as categorical and which were treated as ordinal or continuous.
2. Figure 1A: I think, this figure is a bit of an overkill, because it shows individual data, boxplots and probability density plots. I think that boxplots would contain all information needed. Since the whiskers of box plots are not always defined in the same way, an explanation should be given as to what exactly the whiskers represent in this figure.
3. The fact that the manipulation of the task difficulty did not work as it was supposed to, is only reported in the discussion section. However, statistical material must be reported in the results section, not in the discussion. Therefore, the statistical material reported in lines 292 – 297 on p. 7 must be transferred to the results section. Only its implications must be discussed in the discussion section.
4. p.8, line 320: Although the study by Han et al. (2020) has already been briefly mentioned in the introduction, it would be helpful for readers who are not as familiar with this study as the authors if the content of the study were described and discussed in more detail at this point.
5. p. 8, line 331: You describe that the difficult tasks led to a floor effect. However, regarding figure 1A, it rather seems to me that the tasks with low and (supposedly) middle task difficulty both led to a strong ceiling effect in the paper and computer condition, whereas the difficult task only led to slight skewness of the data, but not to a floor effect. I even suppose that this ceiling effect might have contributed to not finding any difference between paper and computer in the “middle” condition. The ceiling effect should be discussed, the floor effect is not presented, in my opinion.
6. I have detected a missing reference, namely Kintsch (1998). I guess, the reference is Kintsch, W. (1998). Comprehension: A paradigm for cognition. Cambridge University Press. Please check references for completeness.
Comments on the Quality of English Language1. There are a few flaws regarding the english language. Therefore, the language should be checked by a native speaker, e.g.:
a. p. 2, line 73: either “although research has” or “although researchers have”
b. p. 3, line 100: either “an entire passage” or “entire passages”
c. p. 3, line 127: “None of them reported a history of…” instead of “All reported no history…”
d. p. 4, line 160: “A part of the word pairs…”
e. p. 5, line 198: “Each task had a score…”
f. p. 7, line 313: “Consistent with this research…”
Reviewer 2 Report
Comments and Suggestions for Authors
I found this article both interesting and easy to read, albeit, it was a bit short. I have a few small typos detected on the attached copy.
The article has two main flaws.
First, the research questions and hypothesis statements are missing. A good researcher always states the hypotheses and this is crucial when there is hypothesis testing.
Second, the limitations of the study are also missing. What would you change if you were to replicate the study and what would you suggest for further research?
Otherwise, I don't see any major issues with the premise of the study, the study design, or the analysis.

Round 2
Reviewer 2 Report
Comments and Suggestions for Authors
I believe the comments were addressed appropriately and I am satisfied with the changes.